# Signature-Kernel Based Evaluation Metrics for Robust Probabilistic and Tail-Event Forecasting

## Abstract

Standard sample-based metrics like Continuous Ranked Probability Score (CRPS) and Quantile Loss (QL) are frequently used in the evaluation of probabilistic time-series forecasting. However, these metrics fail to capture multivariate correlations or assess the ability to capture distribution tails. These global metrics are insensitive to errors on high-utility events in distributional tails due to over-representation of the distribution's body. To accurately measure distributional fidelity on tail-regions and regions of high utility while still having a proper scoring rule, we introduce a family of censored signature kernel metrics. Our proposed metrics concentrate evaluation of forecasts to a focus region, representing high-utility or distributional tails, by collapsing the body of the distribution to a single pivot. Our benchmarks on time-series foundation models (TSFMs) reveal that while a clear ranking can be formed for distribution capture, there is no clear winner for tasks like systemic load prediction. This indicates that models with a strong ability to capture an overall distribution do not produce forecasts with high downstream utility. To encourage the use of the proposed metrics we open-source the efficient signature kernel (ESK) library. This library facilitates batch computation, with custom triton kernels, achieving a speed-up of up to $3.58\times$ compared to implementations via the popular SigKernel library.

## 1. Introduction

Probabilistic forecasting is of growing importance to real-world applications in a diverse range of fields, such as finance, smart energy, medicine, meteorology, and seismology. Tail-distributions represent rare but high-risk events and thus evaluation metrics must capture the ability to predict them. Despite this, modern evaluation metrics frequently focus on the overall estimation of a distribution, saturating the score and overshadowing tail performance. Furthermore, commonly used evaluation metrics such as the Energy Score (ES) and Continuous Ranked Probability Score (CRPS) are insensitive to autocorrelations, with CRPS also inherently assuming independence between variates.

We propose to overcome these limitations by building upon the work of (de Punder et al., 2026), creating a family of novel, risk-aware censored metrics that focus on the statistical tails and high-risk or high-utility regions of the ground truth distribution. To achieve this, we leverage the signature-kernel Maximum Mean Discrepancy (MMD) (Chevyrev & Oberhauser, 2022) as the backbone of our metrics. Our censoring approach provides a dual use of alignment with downstream utility or tail-regions and effective capture of joint-distribution fidelity, while requiring only samples from the ground truth and learned distributions. The practical impact of this can be seen in Fig. 1, which demonstrates the body over-saturation present in standard metrics.

The key contributions established in this paper are:

- We provide a tail-focused metric defined via the signature kernel to accurately capture a forecaster's ability to predict geometric tails.

- Motivated by practical utility, we further provide Sharpe ratio and asymmetric under-prediction based censored metrics to assess forecasters on regions vital to downstream applications.

- We utilize our censored signature-kernel MMD metrics to evaluate time-series forecasting foundation models, providing insights into benefits of the flow-matching framework in Sundial, and the challenges of tasks such as systemic load prediction.

- We produce an open-source library for efficient batch-wise signature-kernel MMD calculations, using the Goursat PDE kernel trick (Salvi et al., 2021) implemented with custom Triton kernels to achieve a $3.58\times$ speed-up over the standard SigKernel library.

[1]Anonymous Institution, Anonymous City, Anonymous Region, Anonymous Country. Correspondence to: Anonymous Author <anon.email@domain.com>.

Preliminary work. Under review by the International Conference on Machine Learning (ICML). Do not distribute.

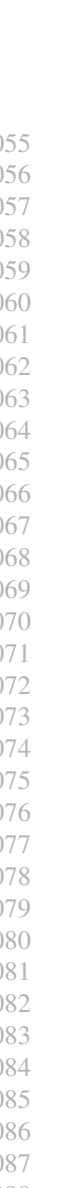
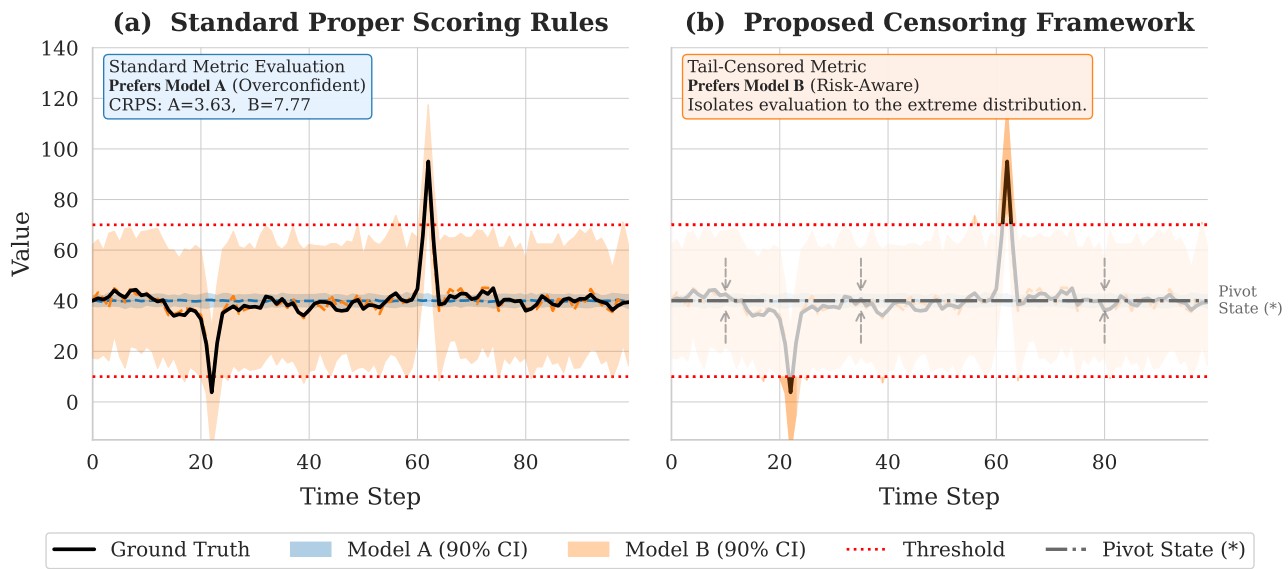

*Figure 1.* **The blindness of standard proper scoring rules compared to our proposed tail-censored framework.** **(a)** Standard metrics are dominated by body forecasting performance. An overconfident forecast (Model A) heavily outperforms a risk-aware forecast (Model B) on CRPS by safely tracking the mean, despite completely missing critical tail events. **(b)** Our proposed metric addresses the over-representation of body forecasts by censoring our metric to only focus on the tail-distribution maintaining strict propriety while correctly preferring the risk-aware model.

Altogether, this work demonstrates the critical need for improved metrics in probabilistic forecasting. It establishes a new class of highly effective tail-distribution and utility aligned metrics which are proper scoring rules even when evaluated on finite samples from a forecaster.

## 2. Problem Formulation

To formally ground our evaluation metrics, we must first define a requirement of any probabilistic forecasting metric: *propriety*. In the context of scoring rules, propriety is the theoretical guarantee that a metric is *proper*, meaning it inherently incentivises a forecaster to report their true belief about an underlying distribution.

---

**Definition 1: Strictly Proper Scoring Rule**

Let $\mathcal{P}$ be a class of probability measures on a $\sigma$-algebra of a sample space $\mathcal{Y}$. A scoring rule $S : \mathcal{P} \times \mathcal{Y} \to \mathbb{R}$ is *proper* relative to $\mathcal{P}$ if the expected score is minimized when the forecasted distribution $Q$ perfectly matches the true distribution $P$:

$$\mathbb{E}_{\mathbf{y} \sim P}[S(P, \mathbf{y})] \leq \mathbb{E}_{\mathbf{y} \sim P}[S(Q, \mathbf{y})] \qquad (1)$$
$$\forall P, Q \in \mathcal{P}.$$

A scoring rule possesses *strict propriety* if this minimum is unique, such that equality holds if and only if $Q = P$.

---

Let $\mathcal{Y} = \mathbb{R}^{D \times h}$ be the sample space of multivariate time-series paths over horizon $h$, equipped with the Borel $\sigma$-algebra $\mathcal{B}(\mathcal{Y})$. The focus of our risk-aware forecasting problem is to evaluate a strictly proper scoring rule exclusively on high-utility or extreme events. Because naively truncating a scoring rule to a localized region violates locally strict propriety, we rely on the framework of *generalized censoring*.

To perform censoring, we introduce a measurable weight function $w : \mathcal{Y} \to [0, 1]$. The weight $w(\boldsymbol{x})$ gives the likelihood that a path $\mathbf{x} \in \mathcal{Y}$ is censored. If $\mathbf{x}$ is censored then it is mapped to a pivot state $*$, else it is left unchanged. So, given a probability measure $P$, using $w$, we map $P$ to a censored measure $P^b$, where the probability of an event $B$ occurring is

$$P^b(B) = \int_B w(\mathbf{x}) P(\mathrm{d}\mathbf{x}) \qquad (2)$$
$$+ \left( \int_{\mathcal{Y}} (1 - w(\mathbf{x})) P(\mathrm{d}\mathbf{x}) \right) \delta_*(B),$$

where $\delta_*$ is the Dirac measure at the pivot $*$.

Censoring, preserves the shape probability mass of paths deemed relevant by $w$, while collapsing censored paths into a single point-mass at the pivot. If $w(\mathbf{x})$ is a binary indicator function, $w(\boldsymbol{x}) = 1$ for $\boldsymbol{x} \in A$ else $w(\boldsymbol{x}) = 0$, we recover *hard* censoring—paths are either always censored or not. However, our formulation of censoring also allows for the use of smooth, differentiable weighting functions.

*Table 1.* **Formal definitions of tail-censored metrics.** Importance functions capture geometric anomalies; utility functions evaluate paths based on downstream actions $(a)$.

| Metric Type | Context & Action (a) | Function Formulation |
|---|---|---|
| ***Importance Functions*** $I(\mathbf{x}_{[t,t+h]})$ | | |
| **Geometric Tails** | Distributional Anomalies | $1 - \frac{2}{M} \sum_{i=1}^{M} k_{\text{norm}}(\mathbf{x}_{[t,t+h]}, z_i) + C_{\text{train}}$ |
| ***Utility Functions*** $U(\mathbf{a}, \mathbf{x}_{[\mathbf{t},\mathbf{t+h}]})$ | | |
| **Sharpe Ratio** | Portfolio Weights $(\mathbf{a} \in \mathbb{R}^D)$ | $\mathbf{a}^\top \hat{\boldsymbol{\mu}} - \frac{\gamma}{2}\mathbf{a}^\top \hat{\boldsymbol{\Sigma}}\mathbf{a}$ |
| **Systemic Load** | Capacity Limit $(a_{t'} \in \mathbb{R})$ | $-\sum_{t'} \big((1-\alpha)(a_{t'} - S_{t'})_+ + \alpha(S_{t'} - a_{t'})_+\big)$ |

*Note:* $S_{t'} = \sum_{d=1}^{D} x_{t',d}$ is aggregate load. Time subscripts $[t, t+h]$ on $\hat{\boldsymbol{\mu}}$ and $\hat{\boldsymbol{\Sigma}}$ omitted for brevity.

## 3. Censored Forecasting Metrics

### 3.1. General Censoring Framework

In practice, probabilistic time-series forecasters output a set of $K$ sample paths over a forecast horizon from time step $t$ to $t+h$. Let us denote a specific path as $\mathbf{x}_{[t,t+h]}$. We characterise a focus region by using a utility measure $U(\mathbf{x}_{[t,t+h]})$ which in turn defines a weight function $w(\mathbf{x}_{[t,t+h]}) \in [0,1]$ used for censoring. A greater utility translates to a larger weight, which dictates the relevance of the path to the downstream task.

This formulation ensures that the probability mass of the empirical samples within the focus region is preserved strictly according to the weighting function, while the irrelevant mass is reassigned to the pivot state. Consequently, the expectation of any measurable function $f$ under the censored measure $P^\flat$ can be expressed as:

$$\mathbb{E}_{P^\flat}[f(X)] = \mathbb{E}_P[w(\mathbf{x}_{[t,t+h]})f(X) \quad (3)$$
$$+ (1 - w(\mathbf{x}_{[t,t+h]}))f(*)].$$

Evaluating a forecasting model strictly on the focus region is equivalent to evaluating the global metric on these censored distributions. Thus, evaluating any strictly proper scoring rule on $P^\flat$ isolates the signal exclusively to the extremes without sacrificing the theoretical guarantees of the underlying metric.

### 3.2. Proposed Censored Metrics

To instantiate our tail-censoring framework for real-world risk profiles, we propose a risk detection mechanism. For a given multivariate time-series path $\mathbf{x}_{[t,t+h]}$, we define a continuous utility score $U(\mathbf{x}_{[t,t+h]})$. To be less sensitive to the choice of what is high utility/risk, we map the utility score to a smoothed censoring weight $w(\mathbf{x}_{[t,t+h]}) \in [0,1]$ using a parametrized logistic sigmoid function:

$$w(\mathbf{x}_{[t,t+h]}) = \sigma\big(\beta(U(\mathbf{x}_{[t,t+h]}) - \tau)\big), \quad (4)$$

where $\beta$ dictates the steepness of the sigmoid and $\tau$ is the critical risk threshold.

We define our pivot state $*$ as the empirical mean path of the training data. This ensures the pivot represents the average path for each application and thus will not fall in the distribution tails.

We apply this censoring scheme to the signature kernel Maximum Mean Discrepancy scoring rule (Sig-MMD), defined in Appendix A. Hence, the general censoring expectation from Eq. 3 is used with respect to the kernel $k_{\text{sig}}$ given by the signature Reproducing Kernel Hilbert Space (RKHS) $\mathcal{H}_{\text{sig}}$. Letting $P$ be the true data distribution and $Q$ be the forecaster's predicted distribution, the final *Censored Signature-Kernel MMD (CSig-MMD)* is defined as the discrepancy between their respective censored mean embeddings:

$$d_{k_{\text{sig}},\flat}^2(P\|Q) = \|\mu_{P^\flat} - \mu_{Q^\flat}\|_{\mathcal{H}_{\text{sig}}}^2$$
$$= \Big\|\mathbb{E}_P[w(\mathbf{x})k_{\text{sig}}(\mathbf{x},\cdot) + (1-w(\mathbf{x}))k_{\text{sig}}(*,\cdot)] \quad (5)$$
$$- \mathbb{E}_Q[w(\mathbf{y})k_{\text{sig}}(\mathbf{y},\cdot) + (1-w(\mathbf{y}))k_{\text{sig}}(*,\cdot)]\Big\|_{\mathcal{H}_{\text{sig}}}^2,$$

where we use $\mathbf{x}$ and $\mathbf{y}$ as shorthand for paths $\mathbf{x}_{[t,t+h]} \sim P$ and $\mathbf{y}_{[t,t+h]} \sim Q$. Here the expectation, $\mathbb{E}_P$, is estimated via Monte-Carlo samples, and censoring is defined on whole paths. This formulation heavily penalizes structural deviations in the focus region while collapsing nominal disagreements to zero distance at the pivot state. A formal summary of the specific utility functions $U(\cdot)$ used to define these focus regions is provided in Tab. 1.

Proof of the strict propriety of our censored metrics can be found in Appendix C.

## 4. Experiments

To empirically validate the discriminatory power of our proposed CSig framework, we benchmarked a diverse suite of state-of-the-art Time Series Foundation Models (TSFMs).

*Table 2.* Aggregated Win/Draw/Loss (W/D/L) outcomes for Time Series Foundation Models across $N = 48$ dataset-horizon combinations. The results demonstrate Sundial's dominance in distributional fidelity (SigMMD), geometric tail forecasting (CSig-Kernel), and systemic load prediction (CSig-Systemic), despite underperforming on financial portfolio tasks. Furthermore, despite disparities in overall distributional accuracy, models exhibited similar performance on the systemic load task, resulting in a higher frequency of draws. Note: MoiraiMoE was excluded from the *electricity* and *traffic* evaluations due to Out-Of-Memory (OOM) failures, resulting in a reduced evaluation count.

| Model | SigMMD (W / D / L) | CSig (Kernel) (W / D / L) | CSig (Systemic) (W / D / L) | CSig (Sharpe) (W / D / L) |
|---|---|---|---|---|
| Chronos | 10 / 0 / 38 | 14 / 0 / 34 | 3 / 6 / 19 | **4** / 0 / 0 |
| Chronos-2 | 14 / 0 / 34 | 11 / 0 / 37 | 4 / 5 / 19 | 0 / 0 / 4 |
| Moirai | 2 / 0 / 46 | 2 / 0 / 46 | 5 / 0 / 23 | 0 / 0 / 4 |
| MoiraiMoE | 0 / 0 / 40 | 0 / 0 / 40 | 0 / 0 / 20 | 0 / 0 / 4 |
| Sundial | **22** / 0 / 26 | **21** / 0 / 27 | **6** / **7** / 15 | 0 / 0 / 4 |
| TimesFM | 0 / 0 / 48 | 0 / 0 / 48 | 3 / 6 / 19 | 0 / 0 / 4 |

The evaluated models encompass a wide array of architectural paradigms, including autoregressive token-based models (Chronos (Ansari et al., 2024), Chronos-2 (Ansari et al., 2025)), masked encoder-decoder patching architectures (Moirai (Woo et al., 2024), MoiraiMoE (Liu et al., 2025b), TimesFM (Das et al., 2024)), and generative flow-matching frameworks (Sundial (Liu et al., 2025c)). The aggregated results, detailed in Table 2, reveal critical insights into how specific inductive biases dictate risk-aware forecasting performance. This win-draw-loss approach is detailed in Appendix B.

**The Dominance of Flow Matching in Continuous Systems.** The benchmarking results demonstrate the overwhelming dominance of Sundial in preserving overall distributional fidelity (SigMMD) and accurately forecasting multivariate anomalies (CSig-Kernel). This empirically validates the theoretical advantage of flow-matching architectures. By natively generating continuous sample trajectories rather than discrete point estimates, flow matchers preserve complex spatio-temporal dependencies (Liu et al., 2025c). This continuous formulation is particularly effective when modelling highly complex physical dynamical systems, as evidenced by Sundial's state-of-the-art performance on the high-dimensional ERA5 atmospheric and climate dataset across all evaluated horizons.

**Performance of Chronos on CSig(Sharpe)** Despite the generalized dominance of flow matching, Chronos wins across horizons on CSig(Sharpe), this is likely due to the presence of exchange in the training set of Chronos, which while used by Sundial represents only 9% of Sundial's training data. This means exchange is a larger proportion of the training data for Chronos which could provide an advantage on this task.

**Validating the Censored Signature-Kernel Framework.** Crucially, these benchmarking results validate the necessity of the proposed CSig framework. The results on the CSig-Systemic task highlight the superiority of our censoring approach. Despite disparities between the models' global distributional accuracy (SigMMD), the models exhibited similar performance on the systemic load task, resulting in a high frequency of "Draws". This is likely due to batches where there were no ground truth samples in the focus region due to low variance in the test set on ETTh1 and ETTm1. When the ground truth has no critical tail events, rather than arbitrarily rewarding one model over another based on irrelevant noise in the distribution body, our censoring framework correctly collapses all non-tail predictions to the pivot state. This results in ties, unless a forecaster predicts a tail event where there is not one, bypassing the body oversaturation that is common in standard scoring rules.

## 5. Conclusions

In this work, we identify the structural blindness of standard proper scoring rules, such as CRPS and Energy Score, to critical tail events due to the phenomenon of body oversaturation. To resolve this, we introduce the Censored Signature-Kernel MMD (CSig-MMD) framework, a novel family of metrics that isolates evaluation strictly to high-risk, utility-aligned regions while mathematically preserving strict local propriety. Our extensive benchmarking of Time Series Foundation Models (TSFMs) using this framework reveals critical architectural insights previously obscured by standard metrics, most notably demonstrating the distinct superiority of continuous flow-matching architectures in preserving complex multivariate correlations. Furthermore, to overcome the inherent computational bottlenecks of evaluating signature kernels at scale, we introduce the open-source ESK library. By utilizing custom Triton kernels to optimize the Goursat PDE kernel trick, we achieve up to a 3.45x speedup, facilitating the tractable, batch-wise evaluation of high-dimensional foundation models. Ultimately, the combination of these theoretical and systems-level contributions establishes a robust, highly scalable foundation for the future of multivariate, risk-aware probabilistic forecasting.

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

# A. Preliminaries

## A.1. Maximum Mean Discrepancy

In this subsection we define proper scoring rules, introduce maximum mean discrepancy, the theoretical underpinning of our proposed scoring rules, and discuss its properties.

**Scoring rules:** A score $S(Q, y)$ is *proper* if the expected score under the true distribution $P$ is minimized by reporting $Q = P$:

$$\mathbb{E}_{y \sim P}[S(P, y)] \leq \mathbb{E}_{y \sim P}[S(Q, y)] \quad \forall Q \in \mathcal{P}.$$

Furthermore, a scoring rule is *strictly proper* if this minimum is unique, such that the equality holds if and only if $Q = P$:

$$\mathbb{E}_{y \sim P}[S(P, y)] = \mathbb{E}_{y \sim P}[S(Q, y)] \iff Q = P.$$

Maximum Mean Discrepancy (Gretton et al., 2012) is a proper scoring rule, and is strictly proper when utilizing a characteristic kernel. A kernel is defined as characteristic if it provides an injective mapping of probability measures into a Reproducing Kernel Hilbert Space (RKHS), thereby allowing for the unique representation and comparison of mean embeddings.

The Maximum Mean Discrepancy is formulated as the squared distance between kernel embeddings in the RKHS:

$$MMD^2(P, Q) = \|\mu_P - \mu_Q\|_{\mathcal{H}}^2,$$

where the kernel embedding $\mu_P \in \mathcal{H}$ is defined as

$$\mu_P := \mathbb{E}_{X \sim P}[k(X, \cdot)].$$

Here, $k$ represents the chosen kernel function and $\mathcal{H}$ is the RKHS forming the codomain of the kernel mapping.

One benefit of the MMD is that it does not require an explicit density function $p(x)$, is a proper scoring rule, and admits both biased and unbiased estimators that preserve properness (Zawadzki & Lahaie, 2015). This is necessary as we do not require forecasters to output a density, we only require samples output from the forecaster to compare with ground truth observations. Additionally, when using a characteristic kernel MMD does not just evaluate the forecast based on means but also on higher order moments embedded in a RKHS. This evaluation of higher order moments provides a more robust method to determine if two samples are from the same distribution.

## A.2. Signature Kernel

The signature of a time series is a graded sequence of statistics that provides a path-based representation derived from the continuous interpolation of discrete data points. A defining property of the signature is its invariance to reparametrisation (Lyons & McLeod, 2025); it remains constant regardless of the sampling frequency or the speed at which the path is traversed, while remaining uniquely sensitive to the sequence order and the geometric profile of the trajectory. Furthermore, specific path augmentations, such as the inclusion of time and base-point coordinates (Morrill et al., 2021), can be applied to ensure the representation captures information regarding translation and temporal duration.

The signature of a continuous path, $x : [a \times b] \to \mathbb{R}^{2d}$ is then defined as:

$$S_{a,t}(x) = \prod_{n=0}^{\infty} S_{a,t}^n(x),$$

where components are defined as:

$$S_{a,t}^0(x) \equiv 1,$$

and for $n > 0$:

$$S_{a,t}^n(x) = \int_{a < t_1 < \ldots < t_n < t} dx_{t_1} \otimes \ldots \otimes dx_{t_n}.$$

As this is an infinite product of iterated integrals, it cannot be computed for Machine Learning usages in this current form, a common solution to this is using a truncated signature of $K$ components. Despite not scaling in complexity with path

length or number of samples, a key drawback of signatures is their scaling with regards to dimensionality of the data as the truncated signature scales exponentially with regards to variates.

Due to their inherent ability to compare sequential data of different length and size, signatures are well-suited to applications within kernel methods for sequential data (Kiraly & Oberhauser, 2019). To this end a kernel can be defined as the scalar product of the signature features as follows:

$$k : BV(\mathcal{H}) \times BV(\mathcal{H}) \to \mathbb{R}, \tag{6}$$

$$s.t. \ k(x.y) \to \langle S(x), S(y) \rangle_{T(\mathcal{H})}, \tag{7}$$

where $BV(\mathcal{H})$ is the space of paths of bounded variation taking values in the Hilbert space $\mathcal{H}$ and $T(\mathcal{H})$ is the tensor algebra of said space. This kernel can be computed without truncation via the kernel trick where it is the solution to a Goursat PDE (Salvi et al., 2021). Using this kernel trick removes the exponential scaling with regards to dimension but has a linear scaling of $\mathcal{O}((L_X + L_Y) \cdot d)$ where $L_X, L_Y$ are the lengths of the input paths of the signature kernel.

## B. Benchmark Results

Here we present the raw results of each time-series foundation model on each proposed metric in the TSLib benchmarking datasets, plus Cloud, ERA5 and EWELD.

These are aggregated in the Win Draw Loss results in Tab. 2.

For each comparison metric $\mu$ (all oriented lower is better) and each applicable (dataset, horizon) pair $(d, h)$, let

$$V_{d,h,\mu} = \{v_m : m \in \mathcal{M}\}, \qquad v^\star = \min_{m \in \mathcal{M}} v_m,$$

where $\mathcal{M}$ is the set of foundation models under comparison. Define a relative draw tolerance

$$\tau = \varepsilon \cdot \max(|v^\star|, 1), \qquad \varepsilon = 10^{-6},$$

and the best set $\mathcal{B}_{d,h,\mu} = \{m : |v_m - v^\star| \leq \tau\}$. The outcome of model $m$ on cell $(d, h, \mu)$ is

$$o_{m,\mu}(d, h) = \begin{cases} \text{Win}, & m \in \mathcal{B} \text{ and } |\mathcal{B}| = 1, \\ \text{Draw}, & m \in \mathcal{B} \text{ and } |\mathcal{B}| \geq 2, \\ \text{Loss}, & m \notin \mathcal{B}. \end{cases}$$

*Table 3.* ERA5: per-model metrics averaged over the 4 forecasting horizons (32,96,192,336), censoring quantile $q = 0.95$.

| Model | CRPS | ES | QL | VS | SigMMD | CSig(K) |
|---|---|---|---|---|---|---|
| Chronos | **12392.2697** | **198796.7311** | 5843.1891 | **1366656.0571** | 0.0780 | 0.0119 |
| Chronos-2 | 17339.3202 | 277219.2407 | **4986.1484** | 1455502.0709 | 0.0808 | 0.0178 |
| Moirai | 13206.9688 | 202844.4186 | 6062.5297 | 3212285.2981 | 0.0804 | 0.0306 |
| MoiraiMoE | 13405.0766 | 208850.7198 | 6220.4207 | 2512615.0216 | 0.1016 | 0.0535 |
| Sundial | 13600.0125 | 214223.8899 | 6443.8836 | 1717223.0048 | **0.0758** | **0.0116** |
| TimesFM | 43379.4402 | 688180.1719 | 21689.7194 | 11402554.2212 | 0.1518 | 0.0220 |

*Table 4.* EWELD: per-model metrics averaged over the 4 forecasting horizons (32,96,192,336), censoring quantile $q = 0.95$.

| Model | CRPS | ES | QL | VS | SigMMD | CSig(K) |
|---|---|---|---|---|---|---|
| Chronos | 2.1039 | 14.0908 | 0.9903 | 142.0499 | 0.0868 | 0.0260 |
| Chronos-2 | 2.7553 | 19.0152 | **0.8449** | **136.3615** | 0.0871 | 0.0254 |
| Moirai | **2.0673** | 14.4468 | 0.9356 | 162.8767 | 0.1191 | 0.0630 |
| MoiraiMoE | 2.1386 | 15.6201 | 0.9782 | 4404.5430 | 0.2454 | 0.1989 |
| Sundial | 2.0936 | **14.0888** | 0.9771 | 138.5356 | **0.0855** | **0.0249** |
| TimesFM | 3.7194 | 24.0518 | 1.8597 | 211.5702 | 0.1432 | 0.0337 |

*Table 5.* Electricity: per-model metrics averaged over the 4 forecasting horizons (32,96,192,336), censoring quantile $q = 0.95$.

| Model | CRPS | ES | QL | VS | SigMMD | CSig(K) | CSig(Sys) |
|---|---|---|---|---|---|---|---|
| Chronos | **28.3884** | **428.3889** | 13.1385 | 11490.5083 | 0.3392 | **0.0594** | 0.0222 |
| Chronos-2 | 41.1392 | 629.1805 | **12.3468** | 11427.7006 | 0.3522 | 0.0807 | 0.0224 |
| Moirai | 37.6551 | 569.8042 | 17.1594 | 18704.1568 | 0.4004 | 0.0857 | 0.0229 |
| MoiraiMoE | — | — | — | — | — | — | — |
| Sundial | 28.5554 | 437.8120 | 13.4734 | **10320.4021** | **0.3368** | 0.0658 | **0.0212** |
| TimesFM | 110.3217 | 1586.6745 | 55.1608 | 54125.3804 | 0.8037 | 0.1351 | 0.0751 |

*Table 6.* Exchange Rate: per-model metrics averaged over the 4 forecasting horizons (32,96,192,336), censoring quantile $q = 0.95$.

| Model | CRPS | ES | QL | VS | SigMMD | CSig(K) | CSig(Shp) |
|---|---|---|---|---|---|---|---|
| Chronos | 0.0212 | 0.0836 | 0.0101 | 0.1380 | 0.0588 | 0.0136 | **0.0069** |
| Chronos-2 | 0.0288 | 0.1125 | **0.0083** | 0.1316 | **0.0539** | **0.0122** | 0.0191 |
| Moirai | 0.0191 | **0.0760** | 0.0088 | **0.1225** | 0.0837 | 0.0458 | 0.0108 |
| MoiraiMoE | 0.0194 | 0.0817 | 0.0086 | 15.1015 | 0.1649 | 0.1275 | 0.0318 |
| Sundial | 0.0229 | 0.0907 | 0.0109 | 0.1312 | 0.0661 | 0.0146 | 0.0498 |
| TimesFM | 0.0653 | 0.3676 | 0.0327 | 1.1925 | 0.1749 | 0.0791 | 0.0347 |

*Table 7.* Illness: per-model metrics averaged over the 4 forecasting horizons (24,32,64,96), censoring quantile $q = 0.95$.

| Model | CRPS | ES | QL | VS | SigMMD | CSig(K) |
|---|---|---|---|---|---|---|
| Chronos | 14396.3311 | 91735.0624 | 6701.4978 | 119478.4044 | 0.2761 | 0.2477 |
| Chronos-2 | **13024.6973** | **81193.0061** | **4470.2262** | **64226.6700** | **0.2758** | **0.2462** |
| Moirai | 17746.7150 | 111197.2272 | 8173.3285 | 193927.5374 | 0.4851 | 0.4630 |
| MoiraiMoE | 15526.8479 | 98150.0155 | 7020.3239 | 149161.6943 | 0.4671 | 0.4466 |
| Sundial | 15247.1711 | 97701.1054 | 7329.4689 | 110321.4784 | 0.2800 | 0.2524 |
| TimesFM | 33356.5265 | 204183.4473 | 16678.2631 | 316798.8141 | 0.4293 | 0.3677 |

*Table 8.* Traffic: per-model metrics averaged over the 4 forecasting horizons (32,96,192,336), censoring quantile $q = 0.95$.

| Model | CRPS | ES | QL | VS | SigMMD | CSig(K) | CSig(Sys) |
|---|---|---|---|---|---|---|---|
| Chronos | 0.0083 | 0.1148 | 0.0040 | 9.2239 | 0.3083 | 0.0677 | 0.0604 |
| Chronos-2 | 0.0103 | 0.1354 | **0.0034** | **8.0413** | **0.2188** | 0.0289 | 0.0553 |
| Moirai | **0.0082** | **0.1034** | 0.0038 | 9.0159 | 0.2220 | **0.0197** | **0.0545** |
| MoiraiMoE | — | — | — | — | — | — | — |
| Sundial | 0.0090 | 0.1112 | 0.0042 | 9.2682 | 0.2350 | 0.0199 | 0.0582 |
| TimesFM | 0.0540 | 0.4685 | 0.0270 | 29.8762 | 0.3414 | 0.0219 | 0.0599 |

*Table 9.* Weather: per-model metrics averaged over the 4 forecasting horizons (32,96,192,336), censoring quantile $q = 0.95$.

| Model | CRPS | ES | QL | VS | SigMMD | CSig(K) |
|---|---|---|---|---|---|---|
| Chronos | 27.2279 | 333.1120 | 12.8625 | 8140.7199 | 0.2030 | 0.1630 |
| Chronos-2 | 22.2794 | 273.6474 | **6.8771** | **3429.8389** | 0.1915 | 0.1521 |
| Moirai | 29.2047 | 364.8068 | 13.2956 | 11462.2183 | 0.3711 | 0.3395 |
| MoiraiMoE | 36.0588 | 521.1375 | 16.9212 | 45322.9101 | 0.6527 | 0.6279 |
| Sundial | **17.1730** | **204.1860** | 8.0838 | 3824.7949 | **0.1858** | **0.1482** |
| TimesFM | 44.3661 | 526.5002 | 22.1831 | 11897.6834 | 0.3568 | 0.2829 |

*Table 10.* ETTh1: per-model metrics averaged over the 4 forecasting horizons (32,96,192,336), censoring quantile $q = 0.80$.

| Model | CRPS | ES | QL | VS | SigMMD | CSig(K) | CSig(Sys) |
|---|---|---|---|---|---|---|---|
| Chronos | 1.3873 | 5.2677 | 0.6666 | 20.3298 | 0.3688 | 0.3330 | 0.0000 |
| Chronos-2 | 1.6708 | 6.3082 | **0.4976** | **15.6039** | **0.3055** | **0.2737** | 0.0000 |
| Moirai | **1.1584** | 4.4127 | 0.5290 | 16.3414 | 0.3336 | 0.3021 | 0.0184 |
| MoiraiMoE | 1.1704 | 4.5399 | 0.5384 | 17.4253 | 0.3617 | 0.3303 | 0.0405 |
| Sundial | 1.1756 | **4.3454** | 0.5515 | 15.6449 | 0.3071 | 0.2755 | **0.0000** |
| TimesFM | 2.7333 | 10.1524 | 1.3666 | 32.5758 | 0.6035 | 0.5324 | **0.0000** |

*Table 11.* ETTh1: per-model metrics averaged over the 4 forecasting horizons (32,96,192,336), censoring quantile $q = 0.95$.

| Model | CRPS | ES | QL | VS | SigMMD | CSig(K) | CSig(Sys) |
|---|---|---|---|---|---|---|---|
| Chronos | 1.3873 | 5.2677 | 0.6666 | 20.3298 | 0.3715 | 0.3216 | 0.0000 |
| Chronos-2 | 1.6708 | 6.3082 | **0.4976** | **15.6039** | **0.3083** | **0.2633** | 0.0000 |
| Moirai | **1.1584** | 4.4127 | 0.5290 | 16.3414 | 0.3365 | 0.2924 | 0.0129 |
| MoiraiMoE | 1.1704 | 4.5399 | 0.5384 | 17.4253 | 0.3646 | 0.3206 | 0.0338 |
| Sundial | 1.1756 | **4.3454** | 0.5515 | 15.6449 | 0.3098 | 0.2650 | **0.0000** |
| TimesFM | 2.7333 | 10.1524 | 1.3666 | 32.5758 | 0.6087 | 0.5058 | **0.0000** |

*Table 12.* ETTh2: per-model metrics averaged over the 4 forecasting horizons (32,96,192,336), censoring quantile $q = 0.80$.

| Model | CRPS | ES | QL | VS | SigMMD | CSig(K) | CSig(Sys) |
|---|---|---|---|---|---|---|---|
| Chronos | 2.2214 | 7.8409 | 1.0665 | 18.5037 | 0.2319 | 0.1744 | **0.0145** |
| Chronos-2 | 2.9407 | 10.5519 | 0.9321 | 18.4243 | 0.2322 | 0.1772 | 0.0152 |
| Moirai | 2.0326 | 7.3027 | 0.9359 | 18.5492 | 0.2769 | 0.2315 | 0.0482 |
| MoiraiMoE | **1.9960** | **7.1817** | **0.9190** | 29.2262 | 0.2710 | 0.2251 | 0.0443 |
| Sundial | 2.1459 | 7.6447 | 1.0172 | **17.4689** | **0.2315** | **0.1736** | 0.0148 |
| TimesFM | 4.0856 | 14.1334 | 2.0428 | 27.3904 | 0.3896 | 0.2837 | 0.0162 |

*Table 13.* ETTh2: per-model metrics averaged over the 4 forecasting horizons (32,96,192,336), censoring quantile $q = 0.95$.

| Model | CRPS | ES | QL | VS | SigMMD | CSig(K) | CSig(Sys) |
|---|---|---|---|---|---|---|---|
| Chronos | 2.2214 | 7.8409 | 1.0665 | 18.5037 | 0.2319 | 0.1572 | **0.0147** |
| Chronos-2 | 2.9407 | 10.5519 | 0.9321 | 18.4243 | 0.2322 | 0.1610 | 0.0150 |
| Moirai | 2.0326 | 7.3027 | 0.9359 | 18.5492 | 0.2769 | 0.2188 | 0.0442 |
| MoiraiMoE | **1.9960** | **7.1817** | **0.9190** | 29.2262 | 0.2710 | 0.2120 | 0.0406 |
| Sundial | 2.1459 | 7.6447 | 1.0172 | **17.4689** | **0.2315** | **0.1564** | 0.0147 |
| TimesFM | 4.0856 | 14.1334 | 2.0428 | 27.3904 | 0.3896 | 0.2537 | 0.0147 |

*Table 14.* ETTm1: per-model metrics averaged over the 4 forecasting horizons (32,96,192,336), censoring quantile $q = 0.80$.

| Model | CRPS | ES | QL | VS | SigMMD | CSig(K) | CSig(Sys) |
|---|---|---|---|---|---|---|---|
| Chronos | 1.2470 | 4.7626 | 0.5938 | 17.9248 | 0.1142 | 0.0763 | 0.0000 |
| Chronos-2 | 1.4178 | 5.3958 | **0.4408** | **13.2569** | 0.1079 | 0.0716 | 0.0000 |
| Moirai | 1.3736 | 5.4307 | 0.6283 | 18.2401 | 0.1945 | 0.1617 | 0.0167 |
| MoiraiMoE | 1.4830 | 6.2457 | 0.6995 | 46.4074 | 0.3867 | 0.3583 | 0.2221 |
| Sundial | **1.0216** | **3.8421** | 0.4797 | 13.4544 | **0.1050** | **0.0699** | **0.0000** |
| TimesFM | 2.2882 | 8.6232 | 1.1441 | 27.1954 | 0.3139 | 0.2505 | 0.0102 |

*Table 15.* ETTm1: per-model metrics averaged over the 4 forecasting horizons (32,96,192,336), censoring quantile $q = 0.95$.

| Model | CRPS | ES | QL | VS | SigMMD | CSig(K) | CSig(Sys) |
|---|---|---|---|---|---|---|---|
| Chronos | 1.2470 | 4.7626 | 0.5938 | 17.9248 | 0.1142 | 0.0718 | 0.0000 |
| Chronos-2 | 1.4178 | 5.3958 | **0.4408** | **13.2569** | 0.1079 | 0.0673 | 0.0000 |
| Moirai | 1.3736 | 5.4307 | 0.6283 | 18.2401 | 0.1945 | 0.1578 | 0.0112 |
| MoiraiMoE | 1.4830 | 6.2457 | 0.6995 | 46.4074 | 0.3867 | 0.3549 | 0.2094 |
| Sundial | **1.0216** | **3.8421** | 0.4797 | 13.4544 | **0.1050** | **0.0656** | **0.0000** |
| TimesFM | 2.2882 | 8.6232 | 1.1441 | 27.1954 | 0.3139 | 0.2417 | 0.0035 |

*Table 16.* ETTm2: per-model metrics averaged over the 4 forecasting horizons (32,96,192,336), censoring quantile $q = 0.80$.

| Model | CRPS | ES | QL | VS | SigMMD | CSig(K) | CSig(Sys) |
|---|---|---|---|---|---|---|---|
| Chronos | 1.6978 | 6.0346 | 0.8086 | 12.0527 | **0.1357** | 0.0962 | 0.0082 |
| Chronos-2 | 2.2096 | 7.9233 | **0.7036** | 11.9174 | 0.1361 | 0.0966 | **0.0080** |
| Moirai | 1.7095 | 6.2744 | 0.7795 | 13.5077 | 0.1905 | 0.1567 | 0.0239 |
| MoiraiMoE | 1.7810 | 6.9059 | 0.8109 | 79.3704 | 0.3070 | 0.2771 | 0.1294 |
| Sundial | **1.6583** | **5.9138** | 0.7828 | **11.3626** | 0.1368 | 0.0974 | 0.0081 |
| TimesFM | 3.1541 | 11.0615 | 1.5771 | 19.0527 | 0.2525 | 0.1779 | 0.0098 |

*Table 17.* ETTm2: per-model metrics averaged over the 4 forecasting horizons (32,96,192,336), censoring quantile $q = 0.95$.

| Model | CRPS | ES | QL | VS | SigMMD | CSig(K) | CSig(Sys) |
|---|---|---|---|---|---|---|---|
| Chronos | 1.6978 | 6.0346 | 0.8086 | 12.0527 | **0.1357** | 0.0883 | **0.0071** |
| Chronos-2 | 2.2096 | 7.9233 | **0.7036** | 11.9174 | 0.1361 | 0.0889 | 0.0072 |
| Moirai | 1.7095 | 6.2744 | 0.7795 | 13.5077 | 0.1905 | 0.1501 | 0.0185 |
| MoiraiMoE | 1.7810 | 6.9059 | 0.8109 | 79.3704 | 0.3070 | 0.2711 | 0.1131 |
| Sundial | **1.6583** | **5.9138** | 0.7828 | **11.3626** | 0.1368 | 0.0895 | 0.0074 |
| TimesFM | 3.1541 | 11.0615 | 1.5771 | 19.0527 | 0.2525 | 0.1628 | 0.0088 |

*Table 18.* Cloud: per-model metrics averaged over the 4 forecasting horizons (32,96,192,336), censoring quantile $q = 0.95$.

| Model | CRPS | ES | QL | VS | SigMMD | CSig(K) | CSig(Sys) |
|---|---|---|---|---|---|---|---|
| Chronos | 15464.3531 | 568410.3743 | 7448.9812 | 24065917.2193 | 0.2351 | 0.1807 | 0.1288 |
| Chronos-2 | 14309.7862 | 553075.3847 | **4563.9561** | **12929998.5920** | 0.2285 | 0.1750 | 0.1254 |
| Moirai | 12721.1829 | 462521.2882 | 5853.7970 | 17502615.9516 | 0.3723 | 0.3286 | 0.2653 |
| MoiraiMoE | 14070.3784 | 502964.3693 | 6439.7919 | 23397926.0446 | 0.5005 | 0.4595 | 0.3998 |
| Sundial | **11795.2532** | **454316.3542** | 5558.4274 | 16252973.7503 | **0.2267** | **0.1742** | **0.1245** |
| TimesFM | 36160.4747 | 1238828.9413 | 18080.2368 | 46752415.6752 | 0.3557 | 0.2427 | 0.1788 |

# C. Proofs of Strict Propriety

To establish the strict local propriety of our Censored Signature-Kernel (CSig-MMD) metrics, we rely on Theorem 2 from de Punder et al. (2026). This theorem dictates that a generalized censored scoring rule maintains strict propriety if and only if its chosen pivot state satisfies the condition of *identifiability*. We formalize this requirement for our framework below:

> **Definition C.1: Identifiability of the Pivot State**
>
> Let $A = \{\mathbf{x} \in \mathcal{Y} : U(\mathbf{x}) \geq \tau\}$ denote the focus region defined by a utility threshold $\tau$. A conditional pivot state $(*)$ satisfies identifiability if it can be unambiguously distinguished from the focus region. Formally, its utility score must be strictly bounded below the threshold of interest:
>
> $$U(*) < \tau \tag{8}$$

In this paper, we universally define the pivot state $(*)$ as the empirical arithmetic mean path of the training data: $\bar{\mathbf{x}} = \frac{1}{N} \sum_{i=1}^{N} \mathbf{x}^{(i)}$. In the following subsections, we establish the identifiability of $\bar{\mathbf{x}}$ across all three proposed risk metrics. In all cases, $\tau$ represents an extreme upper quantile (e.g., the $95^{th}$ percentile) of the training utility distribution, strictly isolating highly anomalous or high-stakes events.

## C.1. Identifiability of the Systemic Load Pivot

The systemic load metric evaluates paths based on a downstream news-vendor cost formulation. As defined in Table 2, the utility of a path $\mathbf{x}$ given a capacity limit action $\mathbf{c}$ is the negative operational cost. To isolate extreme capacity failures, we define the focus region using the risk function, $R(\mathbf{x}) = -U(\mathbf{c}, \mathbf{x})$:

$$R(\mathbf{x}) = \sum_{t'} \left( (1 - \alpha)(c_{t'} - S_{t'})_+ + \alpha(S_{t'} - c_{t'})_+ \right) \tag{9}$$

where $S_{t'} = \sum_{d=1}^{D} x_{t',d}$ is the aggregate load.

The asymmetric pinball loss function, $L(S) = (1 - \alpha)(c - S)_+ + \alpha(S - c)_+$, is strictly convex with respect to the aggregate load $S$. Therefore, by applying Jensen's Inequality, the cost of the empirical mean path is bounded by the expected cost of the individual paths:

$$R(\bar{\mathbf{x}}) = \sum_{t'} L(\bar{S}_{t'}) \leq \frac{1}{N} \sum_{i=1}^{N} \sum_{t'} L(S_{t'}^{(i)}) = \mathbb{E}[R(\mathbf{x})] \tag{10}$$

Because real-world systemic capacity distributions such as server loads or power grids are heavily right-skewed, the expected nominal cost $\mathbb{E}[R(\mathbf{x})]$ is significantly lower than the extreme right-tail threshold $\tau$. Therefore, $R(\bar{\mathbf{x}}) \leq \mathbb{E}[R(\mathbf{x})] < \tau$, mathematically satisfying the identifiability condition.

## C.2. Identifiability of the Geometric Tails Pivot

The geometric tail importance function calculates the squared distance in the signature-kernel RKHS between a test path $\mathbf{x}$ and the empirical training distribution to identify structural anomalies:

$$I_{geom}(\mathbf{x}) = 1 - \frac{2}{M} \sum_{i=1}^{M} k_{norm}(\mathbf{x}, \mathbf{z}_i) + C_{train} \tag{11}$$

In the signature RKHS, geometric distance is heavily driven by path variation, sudden discontinuities, and erratic cross-variate correlations, features characteristic of extreme tail events. By definition, the arithmetic mean path $\bar{\mathbf{x}}$ is a smoothed trajectory where volatility and noise have been averaged out.

Furthermore, because all input trajectories are strictly standardized using Reversible Instance Normalization (RevIN) prior to signature embedding, the mean path resides firmly within the high-density body of the distribution. Consequently, its signature kernel similarity to the training set, $\frac{1}{M} \sum k_{norm}(\bar{\mathbf{x}}, \mathbf{z}_i)$, is highly central. Because the threshold $\tau$ is explicitly calibrated to an extreme upper quantile (e.g., the $80^{th}$ or $95^{th}$ percentile) of training distances to isolate structural outliers, the

importance of the smoothed mean path remains fundamentally bounded within the nominal body. Therefore, $I_{geom}(\bar{\mathbf{x}}) \ll \tau$, mathematically satisfying the identifiability condition.

### C.3. Identifiability of the Sharpe ratio

The Sharpe ratio utility function identifies paths presenting optimal portfolio allocation opportunities, evaluated via the Markowitz formulation given a downstream action of portfolio weights $\mathbf{w} \in \mathbb{R}^D$:

$$U_{sharpe}(\mathbf{x}) = \mathbf{w}^\top \hat{\mu}_{\mathbf{x}} - \frac{\gamma}{2} \mathbf{w}^\top \hat{\Sigma}_{\mathbf{x}} \mathbf{w} \tag{12}$$

where $\hat{\mu}_{\mathbf{x}}$ and $\hat{\Sigma}_{\mathbf{x}}$ represent the intra-path expected returns and covariance.

Using the arithmetic mean path $\bar{\mathbf{x}}$ as the pivot assesses a trajectory where idiosyncratic volatility and extreme return spikes have been neutralized by averaging across all $N$ training paths. This mathematical smoothing collapses the intra-path dynamics of $\bar{\mathbf{x}}$, driving the path's realized variance $\hat{\Sigma}_{\bar{\mathbf{x}}}$ toward zero and reducing its realized return $\hat{\mu}_{\bar{\mathbf{x}}}$ to the baseline "market average" profile.

Extreme tail-events in this context are defined as paths with highly anomalous, positive returns in the direction of the chosen portfolio weights $\mathbf{w}$. Because the threshold $\tau$ strictly isolates these extreme, high-return upper quantiles, the utility of the smoothed market-average mean path naturally falls well below this opportunity threshold: $U_{sharpe}(\bar{\mathbf{x}}) \ll \tau$. Identifiability is therefore preserved.

## D. ERA-5 Setup

The variates chosen for ERA-5 are made up of various different variate types, including:

- **Atmospheric Pressure**

    - *sp*: Surface Pressure (pressure at the Earth's surface)
    - *tp*: Total Precipitation

- **Heat Radiation**

    - *ssrd*: Surface Solar Radiation Downwards (Shortwave radiation)
    - *strd*: Surface Thermal Radiation Downwards (Longwave radiation)

- **Surface & Snow**

    - *skt*: Skin Temperature (temperature of the surface interface)
    - *snowc*: Snow Cover (fractional snow cover on the grid)

- **Soil Temperature (4 Layers)**

    - *stl1*: Soil Temperature Level 1 (0–7 cm)
    - *stl2*: Soil Temperature Level 2 (7–28 cm)
    - *stl3*: Soil Temperature Level 3 (28–100 cm)
    - *stl4*: Soil Temperature Level 4 (100–289 cm)

- **Soil Water Level (4 Layers)**

    - *swvl1*: Volumetric Soil Water Layer 1 (0–7 cm)
    - *swvl2*: Volumetric Soil Water Layer 2 (7–28 cm)
    - *swvl3*: Volumetric Soil Water Layer 3 (28–100 cm)
    - *swvl4*: Volumetric Soil Water Layer 4 (100–289 cm)

- **Near-Surface Temperature**

    - *d2m*: 2m Dewpoint Temperature
    - *t2m*: 2m Air Temperature

- **Wind Level (10m Vectors)**

    - *u10*: 10m U-component of Wind (Eastward)
    - *v10*: 10m V-component of Wind (Northward)

# E. ESK Library & Clock-time

To facilitate the efficient evaluation of massive Time Series Foundation Models (TSFMs), we introduce a custom, hardware-aware implementation of the signature-kernel MMD, designated as the ESK (Efficient Signature Kernel) library.

Standard implementations of the Goursat PDE kernel trick suffer from severe computational bottlenecks, particularly scaling poorly when evaluated across the high-dimensional batches typical of foundation model benchmarking. To resolve this, our ESK library utilizes custom Triton kernels to optimize memory access patterns and parallelize execution on GPU hardware.

**Scaling and Clock-Time Analysis.** We benchmarked the computational efficiency of the ESK library against a standard baseline implementation across two high-dimensional datasets (*exchange_rate* and *weather*). The results, detailed in Table 19, demonstrate a profound performance shift dependent on batch size.

At a batch size of 1, the overhead of the Triton kernel dispatches results in a net slowdown (0.26x speedup). However, as the batch size scales to the configurations actually utilized in foundation model evaluation ($B \geq 16$), the ESK implementation demonstrates massive asymptotic performance gains. At $B = 64$, the ESK library achieves a roughly 3.45x speedup over the baseline, reducing the per-sequence execution time from ~4.02ms down to just 1.16ms. While this has a peak memory this is to be expected due to computing batches instead of a loop through a single item. When compared per-sequence the ESK approach achieves up to a $3.58\times$ speed-up with the base using $0.87\%$ of the memory of our ESK implementation due to their C++ and Cuda implementation.

*Table 19.* Computational scaling of the custom ESK Triton implementation versus the standard baseline. Performance scales asymptotically, reaching a ~3.58x speedup at a batch size of 64.

| Dataset | Batch | ESK (Ours) | | | Baseline | | | Speedup | Mem Ratio |
|---|---|---|---|---|---|---|---|---|---|
| | | Time (ms) | Mem/Seq (MB) | Peak (MB) | Time (ms) | Mem/Seq (MB) | Peak (MB) | | (Base/ESK) |
| Exchange | 1 | 15.03 | 87.45 | 87.45 | 3.94 | 53.94 | 53.94 | 0.26x | 0.61x |
| Exchange | 8 | 2.05 | 77.20 | 617.65 | 4.01 | 53.94 | 53.94 | 1.95x | 0.69x |
| Exchange | 16 | 1.29 | 76.46 | 1223.50 | 4.12 | 53.94 | 53.94 | 3.17x | 0.70x |
| Exchange | 32 | 1.20 | 76.12 | 2435.90 | 3.99 | 53.94 | 53.94 | 3.31x | 0.70x |
| Exchange | 64 | 1.16 | 75.91 | 4858.72 | 4.02 | 53.94 | 53.94 | 3.45x | 0.71x |
| Weather | 1 | 15.74 | 91.26 | 91.26 | 4.28 | 57.68 | 57.68 | 0.27x | 0.63x |
| Weather | 8 | 1.94 | 77.68 | 621.47 | 4.04 | 57.68 | 57.68 | 2.08x | 0.74x |
| Weather | 16 | 1.27 | 76.70 | 1227.32 | 3.98 | 57.68 | 57.68 | 3.12x | 0.75x |
| Weather | 32 | 1.20 | 76.24 | 2439.72 | 4.00 | 57.68 | 57.68 | 3.32x | 0.75x |
| Weather | 64 | 1.16 | 75.97 | 4862.54 | 4.01 | 57.68 | 57.68 | 3.45x | 0.75x |
| Cloud | 1 | 15.78 | 99.78 | 99.78 | 4.14 | 66.19 | 66.19 | 0.26x | 0.66x |
| Cloud | 8 | 3.23 | 78.75 | 629.99 | 4.10 | 66.19 | 66.19 | 1.27x | 0.84x |
| Cloud | 16 | 1.33 | 77.24 | 1235.84 | 4.15 | 66.19 | 66.19 | 3.12x | 0.86x |
| Cloud | 32 | 1.22 | 76.51 | 2448.24 | 4.15 | 66.19 | 66.19 | 3.41x | 0.87x |
| Cloud | 64 | 1.17 | 76.11 | 4871.06 | 4.20 | 66.19 | 66.19 | **3.58x** | 0.87x |

*Table 20.* **Comparison of scoring rules and their theoretical properties.** Our proposed Censored Signature-Kernel MMD (CSig-MMD) is the only metric that simultaneously accounts for multivariate temporal dependencies, maintains strict propriety via the kernel trick, and successfully isolates tail events.

| Metric | Cont. Time | Multivariate Corr. | Time Dep. | Proper Est. | Tail Focus |
|---|---|---|---|---|---|
| QL | ✗ | ✗ | ✗ | ✗ | ✗ |
| CRPS | ✓ | ✗ | ✗ | ✗ | ✗ |
| ES | ✓ | ✓ | ✗ | ✗ | ✗ |
| VS | ✓ | ✓ | ✗ | ✗ | ✗ |
| Sig-MMD | ✓ | ✓ | ✓ | ✓ | ✗ |
| **CSig-MMD (Ours)** | ✓ | ✓ | ✓ | ✓ | ✓ |

# F. Background Work

Currently, the most prevalent metrics used to evaluate probabilistic time-series forecasters are quantile-loss (QL) (Ansari et al., 2025; Shchur et al., 2025; Liu et al., 2025c;a; Kan et al., 2022), continuous ranked probability score (CRPS) (Baron et al., 2025; Dai et al., 2025; Aksu et al., 2024; Zhang et al., 2024; Liu et al., 2025c; Cortes et al., 2025; Rasul et al., 2024; Salinas et al., 2019; Kan et al., 2022; Liu et al., 2025b), energy score (ES) (Kan et al., 2022), and, for measuring variate correlations, the variogram score (VS) (Baron et al., 2025). However, each of these metrics evaluates the global distribution as an average across time-steps, this is insensitive to auto-correlations and tail-distribution errors. Despite their popularity, CRPS and QL are calculated in a univariate manner for each component or time step. This implicitly enforces an assumption of independence of variates or hierarchical forecasting, and it fails to penalize models for poorly capturing autocorrelations, an assumption that rarely holds in real-world data. We demonstrate these failure modes via synthetic experiments in Tab. 21. Furthermore, these vulnerabilities have been theoretically validated for both ES (Scheuerer & Hamill, 2015) and CRPS (Koochali et al., 2022). The properties of each key metric are summarized in Tab. 20, which compares whether they operate over a continuous time axis, assess multivariate forecasts, account for temporal dependencies, maintain propriety on finite samples, and focus on tail events. While threshold-weighted CRPS (Gneiting & Ranjan, 2011) has been proposed as a solution for tail forecasting evaluation, it has been proven (de Punder et al., 2026) that such weighted metrics generally fail to preserve locally strict propriety; additionally, being based on CRPS, they inherently suffer from the same insensitivity to correlation structures.

The signature-kernel maximum mean discrepancy (Sig-MMD), originally proposed in (Chevyrev & Oberhauser, 2022), was recently adapted for evaluating models in spatio-temporal domains (Dodson & Dutta, 2025). However, their work focused strictly on weather data and provided no method of evaluating utility functions or censoring, unlike our proposed metrics. In this paper, we establish the efficacy of Sig-MMD for evaluating the probabilistic forecasting of multivariate time series, offering a thorough analysis of its theoretical benefits and an experimental evaluation across synthetic and real-world datasets. Crucially, we identify that standard Sig-MMD lacks the ability to prioritize forecasters that accurately capture the tails of a distribution, and we propose a novel censored metric to resolve this.

Throughout this paper, we omit comparisons with metrics that require an explicit density formulation, such as negative log-likelihood or tail-focused metrics derived from extreme value theory. This omission is because current forecasters only output quantile values or trajectory samples thus restricting the use of these metrics.

*Table 21.* **Sensitivity to Correlation Structures (5 Seeds).** We evaluate how metrics penalize forecasters that break temporal (No Time), spatial (Indep.), or continuous (Jump) dependencies. Scores are relative multipliers against Ground Truth ($F_1 = 1.00 \times \pm$std). Red highlights perverse incentives or "blind spots" where a broken model is rewarded or not significantly penalized ($\leq 1.01\times$). Only our Censored framework consistently identifies the structural degradation.

| Metric | F1. GT | F2. No Time | F3. Indep. | F4. Jump |
|---|---|---|---|---|
| *Standard Baselines* | | | | |
| ES | 1.00±.01 | 1.00±.01 | 1.02±.01 | 2.93±.08 |
| CRPS | 1.00±.01 | 1.00±.01 | 1.00±.01 | 2.07±.05 |
| QL | 1.00±.01 | 1.00±.01 | 1.00±.01 | 2.18±.05 |
| VS | 1.00±.03 | 1.00±.04 | 1.07±.05 | 1.00±.04 |
| *Kernel-Based Metrics* | | | | |
| RbfMMD | 1.00±.33 | 1.33±.33 | 1.33±.33 | 16.67±.67 |
| SigMMD | 1.00±.05 | 1.06±.06 | 1.08±.05 | 1.88±.13 |
| **Ours (Censored Sig-MMD Framework)** | | | | |
| **CSig (Kernel)** | 1.00±.23 | 1.21±.19 | 1.26±.19 | 5.36±1.89 |
| **CSig (Sharpe)** | 1.00±.26 | 1.03±.23 | 1.03±.23 | 1.32±.29 |
| **CSig (Systemic)** | 1.00±.32 | 1.06±.33 | 1.08±.32 | 2.41±1.19 |

# G. Additional Experiments

## G.1. Correlation Awareness

To effectively evaluate probabilistic forecasters, a metric must be acutely sensitive to both temporal and spatial correlation structures. We demonstrate the failure modes of standard metrics through a synthetic dependency experiment. We construct a ground-truth (GT) multivariate time series sampled from a Gaussian Process with strict spatial and temporal correlations. We then compare the metric scores of the GT against three corrupted forecast distributions: one that drops temporal correlations (No Time), one that assumes complete variate independence (Indep.), and one that introduces discontinuous Bernoulli-distributed arrival jumps (Jump).

As shown in Tab. 21, standard proper scoring rules critically fail this foundational diagnostic. Metrics such as CRPS, QL, and ES perversely reward the forecaster (yielding relative distances $< 1.00\times$) for dropping temporal correlations entirely. In contrast, our proposed CSig-MMD family inherits the robust geometric sensitivity of the signature kernel, consistently and correctly applying a penalty to models that fail to capture the true underlying temporal and spatial dynamics.

Additionally, we observe an important, mathematically sound trade-off: while the standard Sig-MMD strictly enforces spatial correlation penalties (reflecting the loss of spatial correlations with a monotonically increasing score), our censored metrics are slightly less sensitive to nominal, "body" correlation structures. Because our framework deliberately collapses nominal Gaussian paths to the pivot state to isolate tail-events, this behaviour is expected.

## G.2. Utility Experiments

While standard proper scoring rules assess the global distributional fit of a forecaster, they often fail to reflect the model's actual value in downstream decision-making. To validate the practical efficacy of our censoring paradigm, we conduct downstream utility experiments that directly map metric evaluation to real-world risk scenarios. Specifically, we simulate optimal decision-making agents that act upon each TSFM's output forecast at horizon $h = 96$. To ensure our evaluation is robust to diverse operational constraints, we evaluate these decisions across a sweep of risk-aversion profiles $\gamma \in \{0.5, 1.0, 2.0, 5.0\}$ for portfolio allocation, and asymmetric cost penalties $\alpha \in \{0.7, 0.8, 0.9, 0.95\}$ for systemic capacity. The results of these experiments, aggregated across five random seeds, are presented in Tab. 22.

On the Cloud dataset, where we implement a news-vendor capacity selection model to dictate per-time-step actions, standard global metrics (CRPS and ES) fail to select the highest-utility forecaster. Interestingly, alongside our censored metrics, both VS and QL successfully identify the correct forecaster across all parameters and seeds while maintaining a high weighted Kendall rank correlation. However, this apparent competence does not generalise across tasks. When evaluating optimal portfolio weight allocation on the Exchange dataset, these same metrics fail; only our utility-censored metric correctly

*Table 22.* **Optimum Decision-Making Utility.** We evaluate how well metrics identify the model with maximum utility. We report the weighted Kendall rank correlation ($\tau$) and the Top-1 Hit Rate (5 seeds). While specific baseline metrics like QL and VS happen to align well with the Systemic Capacity task, they are not reliable, collapsing entirely on Portfolio Allocation (0% hit rate). Our task-aligned CSig is the only metric that consistently identifies the optimal model across both application scenarios.

| Metric | Exchange (Portfolio Allocation) | | Cloud (Systemic Capacity) | |
|---|---|---|---|---|
| | Weighted $\tau$ ($\uparrow$) | Top-1 Hit Rate | Weighted $\tau$ ($\uparrow$) | Top-1 Hit Rate |
| *Standard Baselines* | | | | |
| CRPS | $+0.55 \pm 0.08$ | 0 / 20 | $+0.15 \pm 0.07$ | 0 / 20 |
| ES | $+0.55 \pm 0.08$ | 0 / 20 | $+0.33 \pm 0.14$ | 0 / 20 |
| QL | $+0.20 \pm 0.12$ | 0 / 20 | $+0.78 \pm 0.05$ | **20 / 20** |
| VS | $-0.26 \pm 0.05$ | 0 / 20 | $\mathbf{+0.92 \pm 0.05}$ | **20 / 20** |
| *Kernel-Based Metrics* | | | | |
| SigMMD | $+0.16 \pm 0.06$ | 0 / 20 | $+0.48 \pm 0.13$ | 16 / 20 |
| **Ours (Censored Framework)** | | | | |
| CSig (Kernel) | $-0.21 \pm 0.07$ | 0 / 20 | $+0.45 \pm 0.05$ | **20 / 20** |
| **CSig (Task-Aligned)** | $\mathbf{+0.85 \pm 0.10}$ | **16 / 20** | $+0.42 \pm 0.07$ | **20 / 20** |

selects the optimal forecaster, achieving a rank correlation of $+0.85$ and a Top-1 hit rate of $80\%$. This demonstrates a fundamental issue: while standard metrics might align with specific tasks, they offer no guarantee of generalisable performance. Conversely, our utility-censored metrics effectively serve a dual purpose, evaluating distributional fidelity within high-utility regions while reliably identifying the forecaster that maximizes downstream utility across domains.

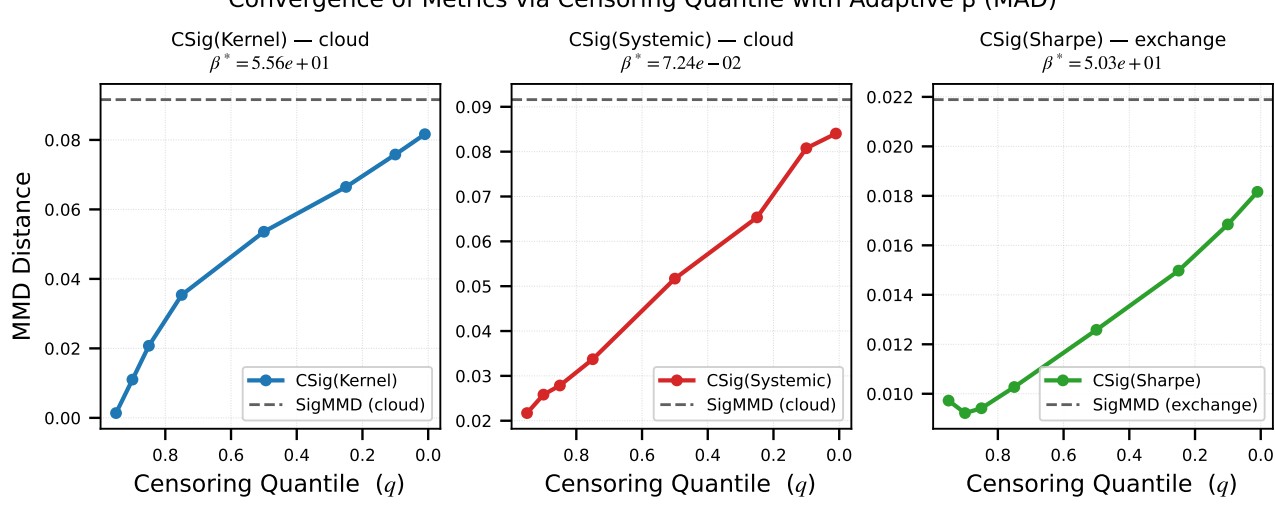

*Figure 2.* **Convergence behaviour of censored metrics under optimal** $\beta^*$**.** As the censoring threshold quantile (x-axis) approaches zero i.e., the focus region expands to encompass the entire dataset, the proposed CSig metrics monotonically converge toward the standard global SigMMD score, illustrated by the dashed straight line. Calibrating the sigmoid steepness parameter $\beta^*$ via the Median Absolute Distance ($\text{MAD}_U$) guarantees this theoretically consistent convergence across distinct datasets and risk definitions.

## H. Ablation Study

We ablate the quantile which sets the censoring threshold $\tau$, how to select $\beta$, and how that affects the convergence behaviour of the proposed censored metrics. For clock-time ablations and a description of the ESK library please see Appendix. E. We propose the setting of $\beta$ based on its similarities to a kernel density estimator, thus the performance of the sigmoid in our case can be optimised, where $\sigma_U^2$ is the variance of the utility function calculated on the ground-truth data. To select the optimum value $\beta^*$, we must consider outliers skewing the data. Thus we utilise Median Absolute Distance (MAD), defined on utility values $\text{MAD}_U$, with a constant to represent the standard deviation of a Gaussian $\sigma_G$. This gives us $\beta^* = \frac{1}{\sigma_G \text{MAD}_U}$. This data based setting of $\beta^*$ causes all metrics in our ablations to exhibit the desired convergence behaviour as the focus region converges to the whole sample set. This behaviour can be seen in Figure. 2.

## I. Practical Advice

**Adjustments for Numerical Stability.** The signature kernel is inherently susceptible to numerical instability, particularly on real-world datasets where un-normalised variance can cause the iterated integrals of the Goursat PDE to scale exponentially, leading to overflow errors. To robustly compute these distances in practice, we implement several critical systems-level mitigations. First, all PDE solver computations within the ESK library are strictly executed using 64-bit floating-point precision (FP64) to prevent precision loss and catastrophic overflow. Second, we apply Reversible Instance Normalization (RevIN) (Kim et al., 2021) to the input paths, standardizing the scale of the trajectories prior to signature embedding without leaking look-ahead information. Finally, we utilize the Laplace kernel as our base static kernel. This explicitly prevents the gradient saturation and distance-collapse issues commonly observed with RBF kernels in high-dimensional trajectory spaces, while avoiding the correlation insensitivity we empirically observed when utilizing Matérn kernels with $\nu \in \{1.5, 2.5\}$.

**When to Use a Censored Metric.** The mathematical propriety of a censored metric is, by design, strictly localized to the censored distribution. While this localization perfectly isolates evaluation to the focus region, neutralizing the body over-saturation of standard metrics, it deliberately blinds the metric to nominal, everyday forecasting performance. Therefore, if a downstream application requires a holistic assessment of a model's capabilities, a censored metric should not be used in isolation. Instead, we advocate for a dual-evaluation paradigm: researchers should utilize an uncensored metric, such as standard Sig-MMD, to verify the general distributional fidelity of the forecaster, alongside CSig-MMD to rigorously evaluate risk-aware tail performance. This paired approach successfully mitigates the risk of failing to distinguish between two models that exhibit identical performance on extreme events but starkly different baselines on nominal data, a rigorous methodology we have adhered to throughout our foundation model benchmarking.

