# OpenReview forum: "Signature-Kernel Based Evaluation Metrics for Robust Probabilistic and Tail-Event Forecasting"
_ICML.cc/2026/Workshop/FMSD — FMSD @ ICML 2026 Poster_

### Official Review · Reviewer_Gaqw · 2026-05-18

**Rating:** 5
**Confidence:** 3

**Review:**

## Summary
The paper proposes a new family of scoring rules for probabilistic time-series forecasting based on the signature kernel. The core idea is to censor the distribution by collapsing the body to a pivot point, allowing the metric to focus evaluation on tail regions or other high-utility regions of the distribution. The authors also provide an efficient Triton-based implementation of the kernel and evaluate several TSFMs using the proposed metric, showing that strong overall distributional performance does not necessarily translate to strong downstream utility.

## Strengths
The observation that existing evaluation metrics may overlook important failure modes is interesting and relevant. In particular, the focus on tail behavior and high-utility events is well motivated in principle. The topic is also well aligned with the workshop theme through the evaluation of TSFMs. The paper additionally includes substantial engineering effort through the efficient Triton implementation.

## Weaknesses
- While the core idea is interesting, the motivation for introducing a new evaluation metric is not yet fully convincing. The paper does not clearly explain why simpler alternatives — such as evaluating QL or CRPS only at extreme quantiles (e.g., 0.90, 0.95, 0.99) — are insufficient. For the motivating examples in Figure 1, these standard approaches already seem adequate.

- The paper also provides little intuition for the proposed metric beyond the mathematical derivation. In contrast, metrics such as CRPS and QL have well-understood interpretations and connections to calibration, interval scores, and coverage (e.g. "Forecast Scoring and Calibration" lecture notes by Tibshirani, 2023). It is therefore unclear when practitioners should prefer the proposed metric over existing approaches.

- The experimental section is difficult to follow. Important details, including the datasets used, are not clearly described in the main paper and instead must be inferred from the appendix. In addition, the evaluation focuses on standard electricity and traffic forecasting benchmarks, where the importance of tail behavior is not obvious. The paper would benefit from concrete downstream applications where tail events are genuinely critical and where standard metrics fail to capture downstream utility (e.g. value at risk).

- Some experimental results, such as Chronos-2 significantly underperforming the original Chronos model are contrary to other large-scale evaluation and might suggest a bug in the implementation (e.g. using bfloat16 instead of float32).

## Justification
The paper contains an interesting idea, and the authors clearly invested significant effort into both the theoretical development and implementation. However, the motivation for introducing a new metric is not yet fully convincing, and the paper lacks clear downstream use cases showing why existing tail-focused evaluations are insufficient. I would encourage resubmission after these concerns are addressed but currently lean towards rejection.

---

### Official Review · Reviewer_yueh · 2026-05-19
**Does this belong elsewhere?**

**Rating:** 3
**Confidence:** 4

**Review:**

Summary:
The paper discusses metrics for probabilistic time series forecasting. It then conducts some experiments on time series foundation models.

Strengths:
Code, the ESK library

Area for improvement:
Increase the relevance for this workshop focused on foundation models for structured data. The current submission is a metrics paper that happens to evaluate on TSFMs; the core contribution (censored signature kernel MMD) is independent of foundation models.

Detailed comments:
The paper is actually about metrics, and the benchmark could have been about any other time series model, not TSFMs. That's why I'm questioning its relevance for this venue. Only the application discusses TSFMs. If you removed the phrase "foundation model" and replaced it with "probabilistic forecaster," none of the technical contributions would change — which tells me this is a metrics paper, not a foundation models paper.

---

### Official Review · Reviewer_QUSd · 2026-05-22
**Review of Signature-Kernel Based Evaluation Metrics for Robust Probabilistic and Tail-Event Forecasting**

**Rating:** 7
**Confidence:** 3

**Review:**

## Summary
The authors note that the focus of traditional probabilistic forecasting metrics (like e.g. CRPS) on the main mass of the pdf, causes them to underestimate the importance of correctly forecasting the distribution tails, which are central for some problems as they correspond to high-utility events (like e.g. systemic load prediction). To tackle this, authors introduce a family of metrics, which use a kernel that (probabilistically) collapses the body of the distribution to a single point ("pivot"), while leaving the tails intact. The authors show that this procedure allows the metrics to be still be proper, which is an important theoretical property for scoring. The authors evaluate several TSFMs over multiple datasets with the proposed metrics, as well as open source an efficient library implementation (ESK, "efficient signature kernels").

## Strengths
* The authors correctly point out to an important weakness of common probabilistic-forecast-evaluation procedures, where the significance of the distribution tails can be heavily under-appreciated.
* The authors make a practical way to evaluate models according to how accurately they predict paths of more "practical importance" (aka utility). In some domains this evaluation has potential to become the new state of the art, especially given that they provide a open-source library implementation
* The authors use their metrics to evaluate multiple models and datasets and show interesting results (e.g. how the model ranking depends on what metric they choose).

## Areas for Improvement and Comments
* I think that section 3 (and Table 1) could benefit from a clearer presentation, it took me multiple reads to understand what is happening.
* I am not a big fan of the win/draw/loss framework to rank models. Something like median rank could be more informative. Also, it would definitely be interesting to have some non-TSFM baselines in the comparison.
* If you have a "wild" fat-tailed distribution (e.g. Cauchy), taking mean as pivot can be risky, I'd expect something like median or winsorized mean to be more robust.
* I'm not sure if the speed up can be measured so accurately as to report it as "up to 3.58", a suspicion confirmed by having 3.45x speed-up reported in the Conclusions. I'd take 3.5 as a coarse average of maximum speed-up over the datasets reported in Table 19.